# Beat perception in polyrhythms: Time is structured in binary units

**Cecilie Møller** [ID] * [◐], **Jan Stupacher** [ID] [◐], **Alexandre Celma-Miralles** [◐], **Peter Vuust**

Center for Music in the Brain, Department of Clinical Medicine, Aarhus University & The Royal Academy of Music Aarhus/Aalborg, Aarhus C, Denmark

◐ These authors contributed equally to this work.
* cecilie@clin.au.dk

**Data Availability Statement:** All data is available at https://researchbox.org/278.

**Funding:** Center for Music in the Brain is funded by the Danish National Research Foundation (DNRF117). Jan Stupacher is supported by an

## Abstract

In everyday life, we group and subdivide time to understand the sensory environment surrounding us. Organizing time in units, such as diurnal rhythms, phrases, and beat patterns, is fundamental to behavior, speech, and music. When listening to music, our perceptual system extracts and nests rhythmic regularities to create a hierarchical metrical structure that enables us to predict the timing of the next events. Foot tapping and head bobbing to musical rhythms are observable evidence of this process. In the special case of polyrhythms, at least two metrical structures compete to become the reference for these temporal regularities, rendering several possible beats with which we can synchronize our movements. While there is general agreement that tempo, pitch, and loudness influence beat perception in polyrhythms, we focused on the yet neglected influence of beat subdivisions, i.e., the least common denominator of a polyrhythm ratio. In three online experiments, 300 participants listened to a range of polyrhythms and tapped their index fingers in time with the perceived beat. The polyrhythms consisted of two simultaneously presented isochronous pulse trains with different ratios (2:3, 2:5, 3:4, 3:5, 4:5, 5:6) and different tempi. For ratios 2:3 and 3:4, we additionally manipulated the pitch of the pulse trains. Results showed a highly robust influence of subdivision grouping on beat perception. This was manifested as a propensity towards beats that are subdivided into two or four equally spaced units, as opposed to beats with three or more complex groupings of subdivisions. Additionally, lower pitched pulse trains were more often perceived as the beat. Our findings suggest that subdivisions, not beats, are the basic unit of beat perception, and that the principle underlying the binary grouping of subdivisions reflects a propensity towards simplicity. This preference for simple grouping is widely applicable to human perception and cognition of time.

## Introduction

In speech, music, and natural environments, we automatically group, subdivide, and structure sound sequences evolving in time. The function of such hierarchical structures is to scaffold and anticipate upcoming auditory events and to facilitate detection of unexpected events through a process that has been termed "predictive timing" [1, 2]. This perceptual grouping of

Erwin Schrödinger fellowship from the Austrian Science Fund (FWF J4288). The funders had no role in study design, data collection and analysis, decision to publish, or preparation of the manuscript.

**Competing interests:** The authors have declared that no competing interests exist.

temporal events is a cognitive mechanism essential for reducing complexity and making sense of the vibrant sensory environment surrounding us.

In search of a universal principle that can explain the ubiquity of rhythms in nature, physiology, attention, speech, poetry, and music, Bolton [3] performed one of the earliest investigations of human rhythm processing. He showed that when listening to unaccented, equally spaced events, such as the isochronous ticks of a clock, listeners tend to subjectively accentuate every fourth or second tick and just rarely accentuate every third tick. Subsequent studies used various paradigms to explore this "tick-tock effect" [4, 5] and its neurophysiological correlates [6, 7]. A general preference for binary or quaternary over ternary grouping is also evident in other tasks involving rhythm perception and production, such as in music [8–12].

The spontaneous clapping, tapping, swaying, and nodding in time with music is a universal human behavior. It provides evidence of our ability to extract and perceive a regular pulse and its underlying hierarchically organized metrical structure. This capacity for beat perception is a fundamental human cognitive skill [13, 14] and present from infancy [15]. Even when listening to complex musical rhythmic structures, which do not accent the beat itself, most people can extract a regular pulse and synchronize their movements to it, indicating that beat perception is a constructive and endogenous process [16].

The regular pulse we emphasize when synchronizing with music represents only one level in a more complex metrical structure. Fig 1 explains the concept of a metrical structure and illustrates its perceptual and behavioral consequences. Fig 1A illustrates how the subdivisions mark the points of the metrical grid, which is established on the basis of the smallest interval between perceptible events of the stimulus. The beat level is the level with which we usually synchronize our body movements. The cycle level marks the onset of the whole repeating pattern. Although the beat level is often the most perceptually salient level, we exhibit high flexibility with regards to synchronizing with any level in the metrical structure [17]. What is considered moving "in time with music" can relate to any level of the metrical structure. Which level of the metrical structure we synchronize our movements with may depend on a number of factors, including stimulus rate, dynamically changing rhythmic accents, and spontaneous motor tempo [18, 19].

In the special case of polyrhythms, two or more metrical structures co-exist and as such, polyrhythms are often used to create tension and increase expressiveness in musical performances. A polyrhythm is created by presenting at least two pulse trains containing coprime numbers of beats within the same periodic cycle, e.g., in ratios of 2:3, 3:4, or 3:5. A listener may perceive one or the other pulse train as representing the underlying beat and extract the corresponding metrical structure. The example depicted in Fig 1 is a 2:3 polyrhythm. Note, that the two manifestations of metrical structure (Fig 1A) are identical at the cycle level as well as at the subdivisions level, which is defined as the least common denominator of the polyrhythm's ratio (i.e., 6 in the case of the 2:3 polyrhythm) What distinguishes the two metrical structures is the beat level, which is defined by the grouping of elements at the subdivision level. On the left of Fig 1A, the ternary subdivision grouping results in two perceived beats per cycle. On the right, the binary subdivision grouping results in three perceived beats per cycle. The resulting metrical structures are organized differently and give rise to distinct and mutually exclusive perceptual experiences depending on how the elements at the subdivision level are grouped (Fig 1B). Two of these perceptual experiences are illustrated with speech examples in Fig 1C.

Previous polyrhythm studies have primarily focused on the constituent pulse trains [20–27], neglecting the polyrhythms' metrical structures and the subdivisions underlying each of the pulse trains. Most of the research has aimed at assessing whether polyrhythms are perceived as integrated or segregated streams [28], while some studies have made efforts to describe the factors that influence beat perception in polyrhythms. Tempo is consistently

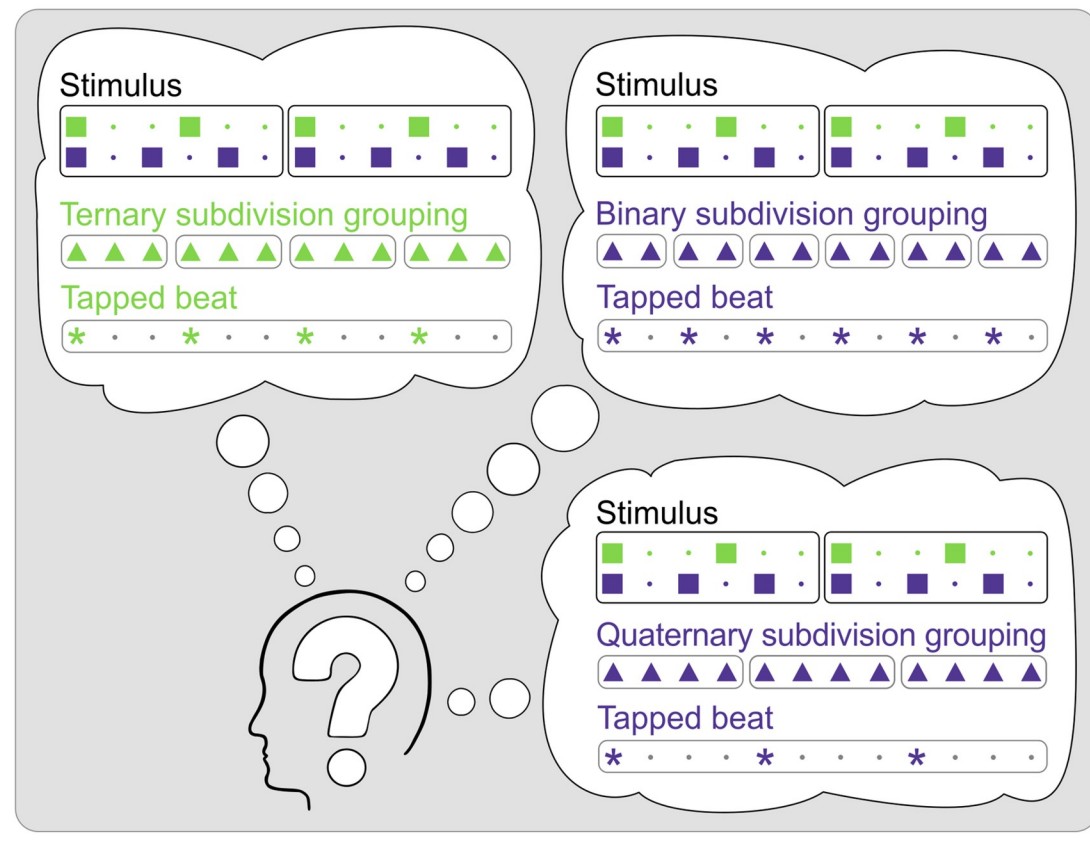

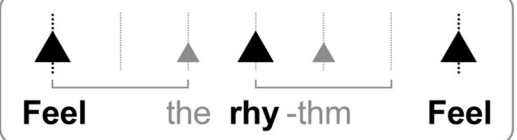

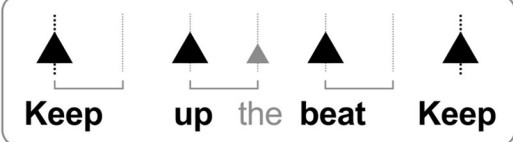

**Fig 1. Examples of metrical structures in rhythms.** A) Left and right panels show the same 2:3 polyrhythm with two different underlying metrical structures, corresponding to the two-beat pulse train with ternary grouped subdivisions (left) and the three-beat pulse train with binary grouped subdivisions (right). B) Three different examples of interpretations of the 2:3 polyrhythm that lead to three different behavioral outcomes when synchronizing body movements—here finger tapping—to the stimulus. The subjective experience of the rhythm's 'feeling' depends on the perceived beat, which in turn depends on the grouping of subdivisions (see Fig I in S1 File, for more information). C) Stressing the bold syllables of the speech examples induces ternary grouped two-beat (left) or binary grouped three-beat (right) interpretations of the 2:3 polyrhythms.

reported to strongly influence whether the faster or the slower pulse train represents the beat, which is also affected by the density, pitch, accentuation of elements and the relative timing between them [29–33]. Importantly, these studies focused on characteristics of the individual pulse trains, taking no notice of the characteristics of the two competing metrical structures that emerge when the pulse trains are superimposed on each other. In order to elucidate how we organize temporal auditory patterns, it is necessary to explicitly consider the hierarchical relationships between metrical levels [34]. Beat perception studies that *do* consider metrical structures tend to focus on the beat and meter levels, e.g., by assessing sensitivity to various manipulations of events at strong and weak beat positions [9, 10, 35]. Yet, beat perception entails perception of subdivisions [34]. Unfolding the empirically established temporal relation between beats and subdivisions, London [34] pointed to the fact that the shortest inter-onset-interval (IOI) that can be perceived as representing beats is approximately 200 ms. In comparison, the shortest IOIs necessary for subjective rhythmization, such as the "tick-tock" effect [3] and for interval discrimination [36] is approximately 100 ms, i.e., corresponding to subdividing the beat by a factor of two. As such, we only perceive a regular beat if the cognitive constraints on temporal perception allow us to perceive the subdivisions of that beat, at least potentially. In a tapping study investigating the benefits and costs of explicitly subdividing the beat, Repp [11] found similar thresholds for motor synchronization rates. Because participants were required to tap only to every second, third, or fourth element of pulse trains presented at different rates, Repp's study also ruled out the possibility that motor constraints influenced participants' ability to make judgements of the quantity of the faster subdivisions. Assuming that successful grouping of subdivisions is necessary for beat perception to occur [34], we have to move our research focus from the beat level to the subdivision level of the metrical structure —especially when assessing beat perception and sensorimotor synchronization in polyrhythms.

To provide a comprehensive account of beat perception that takes into account the most basic level of the metrical hierarchy, the purpose of this online finger tapping study was to assess how subdivision grouping biases listeners to adopt one rather than another possible metrical structure inherent in a given polyrhythm consisting of two pulse trains. Owing to their ambiguous nature, polyrhythms are ideal stimuli for assessing rhythmic interpretations in tapping studies. Such studies rest on the assumption that subjects synchronize their taps with the perceived beat [19, 20]. The paradigm in the present study allowed for categorization of tapping responses into all possible metrical levels, including half and double tempo in relation to the constituent pulse trains (see Fig I in S1 File). Overall, we hypothesized that participants would prefer to tap to a beat with binary rather than ternary or irregular subdivision grouping, and ternary rather than irregular subdivision grouping. This hypothesis was based on the general propensity for binary grouping of isochronous auditory stimuli and their subdivisions [e.g., 3, 7, 37].

Participants were recruited worldwide via social media. In three separate online experiments we manipulated tempo (N = 100), ratio (N = 120), and pitch (N = 80) of pulse trains in various polyrhythms. Tempo was manipulated to assess transition points of tapping preference

within and between metrical structures. Ratio was manipulated to investigate different types of subdivision grouping (binary, ternary, irregular) in the slow and the fast pulse trains at different tempi. Pitch manipulations allowed assessing the effect of low-frequency notes on beat perception relative to the effect of subdivision grouping. We report the results of preregistered main analyses (https://aspredicted.org/yi5si.pdf).

## Method

### Participants

The study included data of 300 participants (159 female, age range 18–75 years, $M$ = 31 years, $IQR$ = 10.5 years). Additional incomplete or duplicate responses were excluded. The majority of respondents grew up in Denmark (32.7%) followed by Spain (12.3%), UK (7.3%), Germany (5.3%), and the US (4.7%). The remaining 37.7% of participants grew up in forty-four different countries. Musicianship was assessed with one item from Ollen's Musical Sophistication Index [38]. Eleven percent considered themselves nonmusicians, 29%, music-loving nonmusicians, 24% amateur musicians, 18% serious amateur musicians, 11% semi-professional musicians, and 8% professional musicians. Participants were randomly assigned to complete either the Tempo ($N$ = 100), Ratio ($N$ = 120) or Pitch ($N$ = 80) Experiment (Fig 2). Sample sizes were determined a priori based on pilot studies and preregistered. Participants were informed that their data would be used for scientific purposes. They were not offered any kind of payment for their participation. The study was conducted in accordance with the guidelines from the

### Tempo Experiment ($N$ = 100)

Cycle tapping for very fast tempi?

Binary subdivision grouping for mid tempi?

Subdivision tapping for very slow tempi?

### Ratio Experiment ($N$ = 120)

Do we have a preference for binary over ternary and ternary over irregular grouping of subdivisions?

**Expected preferences:**

2:3  2:5  3:4  3:5  4:5  5:6

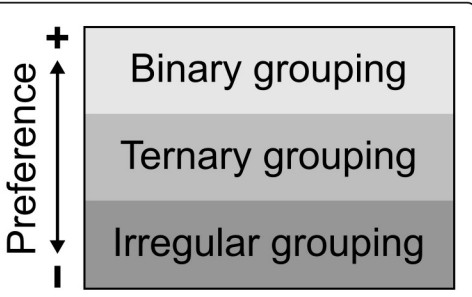

### Pitch Experiment ($N$ = 80)

Do we have a preference for synchronizing with the pulse train containing lower pitched tones?

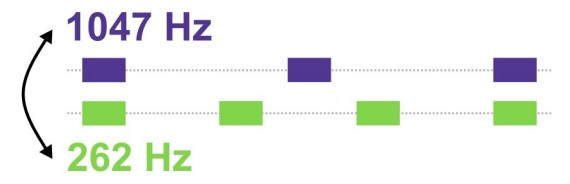

**Fig 2. Experimental overview and main research questions for the three individual experiments.**

Declaration of Helsinki and the Danish Code of Conduct for Research Integrity and Aarhus University's Policy for research integrity, freedom of research and responsible conduct of research. In Denmark, research that does not collect nor store personally identifiable or sensitive information are exempt from IRB approval, which we confirmed in correspondence with the local IRB. All collected data included no personally identifiable information.

## Procedure

Participants were recruited through social media and directed to a webpage containing a Shiny app, which was developed using the JavaScript library jsPsych [39] embedded into psychTestR [40]. Headphones and touch screens were recommended, though the experiment could also be run on computers using internal speakers. After initial assessment and testing of their devices, participants were randomly assigned to the three different experiments, Tempo, Ratio, or Pitch. For the Pitch dataset, we only included participants wearing headphones to ensure a proper representation of low-frequency tones. After a spontaneous motor tempo assessment, which familiarized participants with tapping on their device (touchscreen, touch pad, or mouse), the experimental tapping task was explained. Participants' task was to listen to the polyrhythm and to start tapping with the index finger of their dominant hand when they could clearly "feel" the beat. They continued tapping until the sound stopped. Stimuli were presented once and in random order. After each trial, a 100-point slider allowed participants to rate how difficult it was to find the beat. Finally, participants filled out a short questionnaire assessing their musical and linguistic background. Questionnaire data, spontaneous motor tempo, and difficulty ratings were not analyzed in the present work, which focuses specifically on reporting the tapping data as specified in the preregistration (https://aspredicted.org/yi5si.pdf).

## Finger tapping analyses

We obtained a high proportion of tapping responses. Only 1.3% of the tapping trials were missing in Tempo, 2.0% in Ratio, and 1.0% in Pitch. To remove involuntary double taps and device artefacts, we calculated all the time intervals between consecutive taps (Inter Tapping Intervals, ITIs) and removed the second tap of each ITI shorter than 150 ms (see Fig II(A) in S1 File). We additionally removed the first two taps of each trial. If less than five taps remained, we removed the trial from the analysis. The resulting percentages of excluded trials in this step were 3.7% in Tempo, 4.6% in Ratio, and 3.0% in Pitch.

The timing of the taps was converted into angular measures, i.e., phase in radians. This means that the taps are measured as a circular angle in relation to the timing of a periodic reference event at angle 0. The periodic references were defined as the different metrical levels in each polyrhythm: *cycle*, *slow pulse train*, *slow pulse train—double tempo*, *slow pulse train—half tempo*, *fast pulse train*, *fast pulse train—double tempo*, *fast pulse train—half tempo*, and the common *subdivision* level (see Fig II(B) in S1 File). This method allowed us to obtain the consistency of the taps at each metrical level, even if the device or the participant missed some taps. Because every operating system, browser and tapping device has different delays, we did not analyze stimulus-tapping phase but used ITIs and circular statistics.

In the circular statistics analyses [41], we computed the mean resultant vector of the tapping responses to each stimulus for every metrical level. The tapping consistency is reflected by the length of the mean vector ranging from 0 to 1. To determine the metrical level tapped by the participant in each trial, we took the longest vector length among all the metrical levels and checked whether the taps were uniformly distributed using the Rao's Spacing test and the Rayleigh test. Only when both tests were significant ($p \leq .05$), the tapping responses were assigned to a metrical category. This procedure filtered out non-regular tapping responses, including

irregular grouping of subdivisions and synchronization with the rhythm itself. The percentages of trials not assigned to any metrical category (i.e. non-significant at least in one of the two circular tests) were 12.3% in Tempo, 24.7% in Ratio, and 5.1% in Pitch. The larger proportion of uncategorized responses in the Ratio Experiment reflects the increased complexity of the stimuli presented here. Finally, we confirmed the selection of the categorized metrical level by only accepting those metrical tapping responses (i) whose mean of the ITIs fell in the range of ±15% of the inter-onset interval (IOI), i.e., the tempo in milliseconds, of the categorized meter and (ii) whose standard deviation of the ITIs was smaller than 66% of the IOI. These means and standard deviations were obtained after removing ITIs that fell beyond two standard deviations from the mean ITI of each trial. The percentages of trials rejected in this step were 16.6% in Tempo, 14.7% in Ratio, and 14.7% in Pitch. The combination of linear and circular analyses resulted in the final inclusion of the following percentages of trials with consistent tapping at one of the metrical levels: 66.2% in Tempo, 54.1% in Ratio, and 76.2% in Pitch. See Fig III in S1 File, for visualization of trials excluded in each of the data cleaning steps.

## Tempo experiment

The perception of a regular beat in isochronous sequences of sounds is possible if the tempo is within a range of approximately 30–300 BPM / 2000–200 ms [18, 34, 42]. Particularly salient pulses are perceived at tempi between 80–160 BPM / 750–375 ms, corresponding to our preferred spontaneous motor tempo [4, 43, 44]. Previous studies suggested that, when synchronizing with polyrhythms, individuals tap in time with the pulse train closest to the human preferred tempo, i.e., the faster pulse train in slower tempi and vice versa [32, 33]. In contrast, we expected that individuals synchronize with the pulse train that can be subdivided into binary groups. We assessed 2:3 and 3:4 polyrhythms in a wide range of tempi. The fast pulse train in the 2:3 polyrhythm (i.e., 3) admits binary subdivision whereas the slow pulse train in the 3:4 polyrhythm (i.e., 3) admits binary subdivision.

**Hypotheses.** We expected that any difference in distribution of tapping preference between the two polyrhythms can be explained by differences in subdivision grouping and not by the relative timings of the slow and the fast pulse trains. At moderate tempi, we expected taps to occur in time with the pulse train in which subdivisions could be grouped in pairs (binary). At faster tempi, we expected the pulse train itself to be perceived as subdivisions—again with preferences for binary grouping of subdivisions. At extremely slow and extremely fast tempi, we expected taps to shift towards the subdivisions and the cycle, respectively.

**Stimuli.** All stimuli in this study were created with Ableton Live 8 (Ableton, Berlin, Germany; audio files in https://researchbox.org/278). The Tempo stimuli consisted of 2:3 and 3:4 polyrhythms ranging from very slow (approx. 40 BPM) to very fast tempi (approx. 450 BPM; see Table 1). The two pulse trains in each of the polyrhythms were presented with the same cowbell sound and the same amplitude. The duration of the 15 stimuli was between 18 and 28 s, depending on the ratio and tempo. Every stimulus was presented once in random order and consisted of at least six repetitions of a whole polyrhythm cycle. Additional stimuli for control analyses are shown in Figs IV and V in S1 File.

**Statistical analyses.** Tapping responses were categorized as one of the following metrical categories: *cycle*, *slow pulse train*, double and half tempo of the *slow pulse train*, *fast pulse train*, double and half tempo of the *fast pulse train*, and the common *subdivisions*. Unclear tapping performances that could not be categorized were not analyzed. Within each metrical level, Cochran's Q tests were used to investigate the effect of tempo. McNemar's tests were used to analyze differences between neighboring tempo pairs. All analyses were Bonferroni-corrected for multiple comparisons.

**Table 1. The 15 stimuli of the tempo experiment.**

| 2:3 Polyrhythm | |
| --- | --- |
| **Tempo in ms** | **Tempo in BPM** |
| 1500:1000 | 40:60 |
| 1000:667 | 60:90 |
| 667:444 | 90:135 |
| 444:296 | 135:202 |
| 296:198 | 202:304 |
| 198:132 | 304:456 |
| **3:4 Polyrhythm** | |
| **Tempo in ms** | **Tempo in BPM** |
| 1579:1184 | 38:51 |
| 1186:889 | 51:67 |
| 889:667 | 67:90 |
| 667:500 | 90:120 |
| 500:375 | 120:160 |
| 375:280 | 160:214 |
| 280:211 | 214:284 |
| 211:158 | 284:379 |
| 158:119 | 379:506 |

## Ratio experiment

Although the literature acknowledges that rhythmic interpretation depends on the configuration and in turn on the structure of the polyrhythm [29, 30, 45], no efforts have yet been made to directly assess which particular aspects of the polyrhythm configurations drive tapping preference. In the Ratio Experiment, we assessed preference for metrical structure by focusing on the polyrhythm subdivision level, rather than preference for the constituent pulse trains, as in previous studies [29–31, 33]. The paradigm included polyrhythms ranging from simple (e.g., 2:3) to complex (e.g., 5:6; see Table I in S1 File, for a definition of complexity). Different configurations of polyrhythms give rise to different possible subdivision groupings (binary, ternary, irregular). For example, in a 2:3 polyrhythm, a binary subdivision grouping subserves the three-beat and a ternary subdivision grouping subserves the two-beat (Fig 1A). Accordingly, with respect to subdivision grouping, the 2:3 polyrhythm can be denoted ternary:binary, while for instance the 2:5 polyrhythm can be denoted irregular:binary. Because six subdivisions may be grouped in either two or three, the 5:6 polyrhythm should be denoted binary/ternary:irregular.

**Hypothesis.** We expected that the metrical structure containing simpler subdivision grouping would be preferred over those containing more complex subdivision grouping. This means that we expected the following preferences: Binary grouping (2 or 4) is preferred over ternary grouping (3), and ternary grouping (3) is preferred over irregular grouping (5).

**Stimuli.** The Ratio stimuli consisted of 2:3, 2:5, 3:4, 3:5, 4:5, and 5:6 polyrhythms (audio files in https://researchbox.org/278). The two pulse trains in each polyrhythm were presented with the same cowbell sound and the same amplitude. The tempo of the polyrhythms were based on the duration of their subdivisions, i.e., their least common denominator (Table 2). Two 3:4 polyrhythms with the subdivision tempi 167 and 125 ms, corresponding to pulse train tempi of 90:120 and 120:160 BPM (667:500 and 500:375 ms), were used as anchors. To make the tempo of the pulse trains comparable across the different ratios, 2:3 and 2:5 polyrhythms were additionally slowed down to half tempo, whereas 3:5, 4:5, and 5:6 polyrhythms were

Table 2. The 22 stimuli of the ratio experiment.

| Ratio | Duration of subdivisions in ms | Duration of cycle in ms | Tempo of pulse trains in ms | Tempo of pulse trains in BPM |
|---|---|---|---|---|
| 2:3 | 333, 250, 167, 125 | 2000, 1500, 1000, 750 | 1000:667, 750:500, 500:333, 375:250 | 60:90, 80:120, 120:180, 160:240 |
| 2:5 | 333, 250, 167, 125 | 3330, 2500, 1670, 1250 | 1667:667, 1250:500, 833:333, 625:250 | 36:90, 48:120, 72:180, 96:240 |
| 3:4 | 167, 125 | 2000, 1500 | 667:500, 500:375 | 90:120,120:160 |
| 3:5 | 167, 125, 84, 63 | 2500, 1875, 1260, 945 | 833:500, 625:375, 420:252, 316:189 | 72:120, 96:160, 143:238, 190:317 |
| 4:5 | 167, 125, 84, 63 | 3340, 2500, 1680, 1260 | 833:667, 625:500, 420:335, 316:252 | 72:90, 96:120, 143:179, 190:238 |
| 5:6 | 167, 125, 84, 63 | 5010, 3750, 2520, 1890 | 1000:833, 750:625, 504:420, 377:316 | 60:72, 80:96, 119:143,159:190 |

additionally speeded up to double tempo (Table 2). Following the temporal constraints on beat perception described by London [34] and Repp [11], it is reasonable to assume that beat perception is only possible when the tempo allows for grouping of subdivisions. Consequently, in a metrical structure containing groupings of a large number of subdivisions, the tempo of the pulse train must be slowed down to allow beat perception to occur and to make balanced comparisons between different ratios possible. In other words, comparisons should be made between subdivision tempi, not pulse train tempi. The duration of the 22 stimuli was between 15 and 24 s, depending on the ratio and tempo. Every stimulus consisted of at least five repetitions of a whole polyrhythm cycle.

**Statistical analyses.** Tapping responses were coded as 1 when falling into one of the *fast pulse train* categories (*fast pulse train*, *fast pulse train—double tempo*, or *fast pulse train—half tempo*) and coded as 0 when falling into one of the *slow pulse train* categories (*slow pulse train*, *slow pulse train—double tempo*, or *slow pulse train—half tempo*). These values were averaged across all tempi in each of the polyrhythm ratios (Fig 5A). For statistical analysis, we computed two means: 1) the mean of all polyrhythm ratios in which the slow:fast pulse train relation admits ternary:binary subdivision (2:3), irregular:binary subdivision (2:5, 4:5) or irregular:ternary subdivision (3:5), i.e., ratios with simpler subdivision grouping in the faster pulse train, and 2) the mean of all polyrhythm ratios in which the slow:fast pulse train relation admits binary:ternary subdivision (3:4) or binary/ternary:irregular subdivision (5:6), i.e., ratios with simpler subdivision grouping in the slower pulse train. These two means were compared using a paired sample Wilcoxon test.

## Pitch experiment

The pitch of elements in a musical rhythm is an important factor for beat perception. Low-pitched rhythmic elements increase the sensitivity to timing variation on behavioral and neural levels [46], and EEG activity at meter-related frequencies increases with low-pitch sounds [47]. In general, high energy in bass frequencies are important for inducing movements, such as tapping in time with the beat and dancing [48–51]. When investigating the effect of pitch on beat perception in polyrhythms, Handel and Oshinsky [31] found that participants tended to perceive the lower pitched pulse train as the beat and that this preference counteracted preferences related to the timing of the pulse trains. Here, we investigated this effect of lower pitch in more detail by not only varying pitch, but also loudness between the slow and fast pulse trains in 2:3 and 3:4 polyrhythms. This manipulation was informed by pilot studies showing that without a loudness manipulation the vast majority of participants tap in time with the pulse train admitting binary subdivision grouping. The resulting design allowed us to assess whether preferences for binary subdivision grouping have stronger effects on beat perception than loudness or bass frequencies.

**Hypotheses.** We expected participants' tapping responses to reflect a preference for lower pitched pulse trains. This means that the preference for binary subdivision grouping should be

strengthened when coinciding with the low-pitched pulse train and weakened when coinciding with the high-pitched pulse train.

**Stimuli.** The Pitch stimuli consisted of 2:3 and 3:4 polyrhythms created with marimba sounds at the tempi of 90:135 BPM (667:444 ms) and 90:120 BPM (667:500 ms), respectively (audio files in https://researchbox.org/278). Table 3 details the *pitch* and *loudness* manipulations. In each polyrhythm, one of the pulse trains was pitched low with a peak frequency of 262 Hz (C4) and the other pulse train was pitched higher with a peak frequency of 1047 Hz (C6). This manipulation was counterbalanced. The loudness of the two pulse trains was either the same, moderately louder for the pulse train with ternary subdivisions, or markedly louder for the pulse train with ternary subdivisions. Loudness was measured in Loudness K-weighted Full Scale (LKFS) with the Orban Loudness Meter (version 2.9.6; www.orban.com/meter/). The duration of the 12 stimuli was 17 and 18 s for ratios 2:3 and 3:4, respectively. To exclusively investigate the effect of amplitude, we presented six additional control stimuli with the same loudness manipulation as the experimental stimuli but using the same pitch in both pulse trains (C5 with a peak frequency of 524 Hz; Fig VI in S1 File).

**Statistical analyses.** The dependent variable was defined as the tapping consistency related to the *slow pulse train* minus the tapping consistency related to the *fast pulse train* in each trial. As the tapping consistency is defined as the vector length in circular statistics, this procedure resulted in a value ranging between -1 and 1. Values close to -1 indicate that participants consistently tapped in time with the *slow pulse train*, whereas values close to 1 indicate that participants consistently tapped in time with the *fast pulse train*. Data for the individual factor combinations (2 *pitch* × 3 *loudness* factor levels) were not normally distributed (Shapiro-Wilk p-values < .001). In the preregistration for this study, we planned to use linear mixed effects models for analyzing the pitch data. However, the residuals of these models were not normally distributed, as indicated by visual inspections of Q-Q plots and Shapiro-Wilk tests (all p-values < .001). Consequently, we used two Wilcoxon tests for paired samples to

**Table 3. The 12 pitch stimuli.**

| 2:3 Polyrhythm (90:135 BPM) | | | |
|---|---|---|---|
| **2 pitch** | **3 pitch** | **2 amplitude** | **3 amplitude** |
| low | high | = | = |
| low | high | + 3 LKFS | - 3 LKFS |
| low | high | + 6 LKFS | - 6 LKFS |
| high | low | = | = |
| high | low | + 3 LKFS | - 3 LKFS |
| high | low | + 6 LKFS | - 6 LKFS |
| **3:4 Polyrhythm (90:120 BPM)** | | | |
| **3 pitch** | **4 pitch** | **3 amplitude** | **4 amplitude** |
| low | high | = | = |
| low | high | - 3 LKFS | + 3 LKFS |
| low | high | - 6 LKFS | + 6 LKFS |
| high | low | = | = |
| high | low | - 3 LKFS | + 3 LKFS |
| high | low | - 6 LKFS | + 6 LKFS |

*Note*. Low pitch refers to C4 (peak frequency 262 Hz) and high pitch to C6 (peak frequency 1047 Hz). In the amplitude columns, "=" refers to equal loudness measured in LKFS; +3/+6 LKFS mark increases of the original loudness by 3 and 6 LKFS, respectively; -3/-6 LKFS mark decreases of the original loudness by 3 and 6 LKFS, respectively.

investigate the effect of *pitch* (low pitch in *slow* vs. *fast pulse train*) and two Kruskal-Wallis tests to investigate the effect of *loudness* in 2:3 and 3:4 polyrhythms, separately. We report the results of the nonparametric tests, which were supported by a double check analysis using the linear mixed effects models.

## Results and discussion

In the following section, we report and discuss the results of the three experiments Tempo, Ratio, and Pitch separately, to unravel their individual effects on beat perception in poly-rhythms. In the subsequent General Discussion, we focus on the converging evidence for a preference for binary grouping of subdivisions in musical rhythms and broaden the perspective to binary grouping of temporal units in the perception of time in everyday life.

### Tempo experiment

Consistent with our hypothesis, more participants' responses fell into one of the response categories with binary compared to ternary subdivision groupings (Fig 3). The pulse trains admitting binary subdivision grouping (henceforth "binary"), i.e., *fast pulse train* in 2:3 and *slow pulse train* in 3:4 polyrhythms, showed statistically significant differences in the proportion of responses to the different tempi ($X^2(5) = 128.50$, $p < .001$, $\eta^2_Q = 0.22$, and $X^2(8) = 251.32$, $p < .001$, $\eta^2_Q = 0.28$, respectively; Fig 4). For the pulse trains admitting ternary subdivision grouping (henceforth "ternary"), i.e., *slow pulse train* in 2:3 and *fast pulse train* in 3:4 polyrhythm, no statistically significant differences in the proportion of responses to the different tempi of the stimuli were found (both p-values > .5). In any one of the six tempi in 2:3 and 9 tempi in 3:4 polyrhythms, less than 6% of participants tapped in time with the pulse train with ternary subdivision grouping.

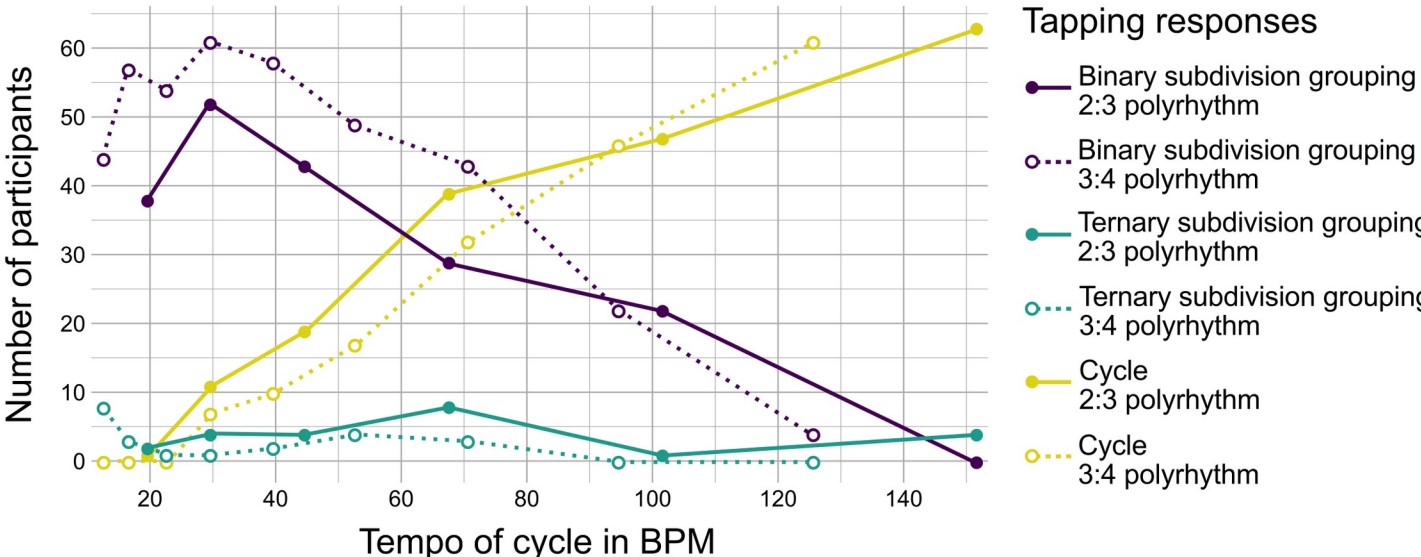

**Fig 3. Participants' preference for subdivision grouping in the tempo experiment.** The different lines in the plot represent responses with binary (dark purple) or ternary (green) subdivision grouping. At slow and moderate tempi, most participants tapped in time with one of the response categories that admitted a binary subdivision regardless of tempo. At faster tempi, participants switched to tapping in time with the cycle (bright yellow). Only a few participants' tapping behavior corresponded to ternary subdivision grouping. Binary subdivision in 2:3 include *fast pulse train* and *fast pulse train—half tempo*. Binary subdivision in 3:4 include *slow pulse train*, *slow pulse train—half tempo*, *slow pulse train—double tempo*, and *fast pulse train—half tempo*. Ternary subdivision in 2:3 include *slow pulse train* and *slow pulse train- double tempo*. Ternary subdivision in 3:4 include *fast pulse train* and *fast pulse train—double tempo*. For an overview of binary and ternary metrical categories, see also the color-coded tapping responses in Fig 4. Note that subdivision grouping is indistinguishable at the *all subdivisions* (not shown) and *cycle* levels.

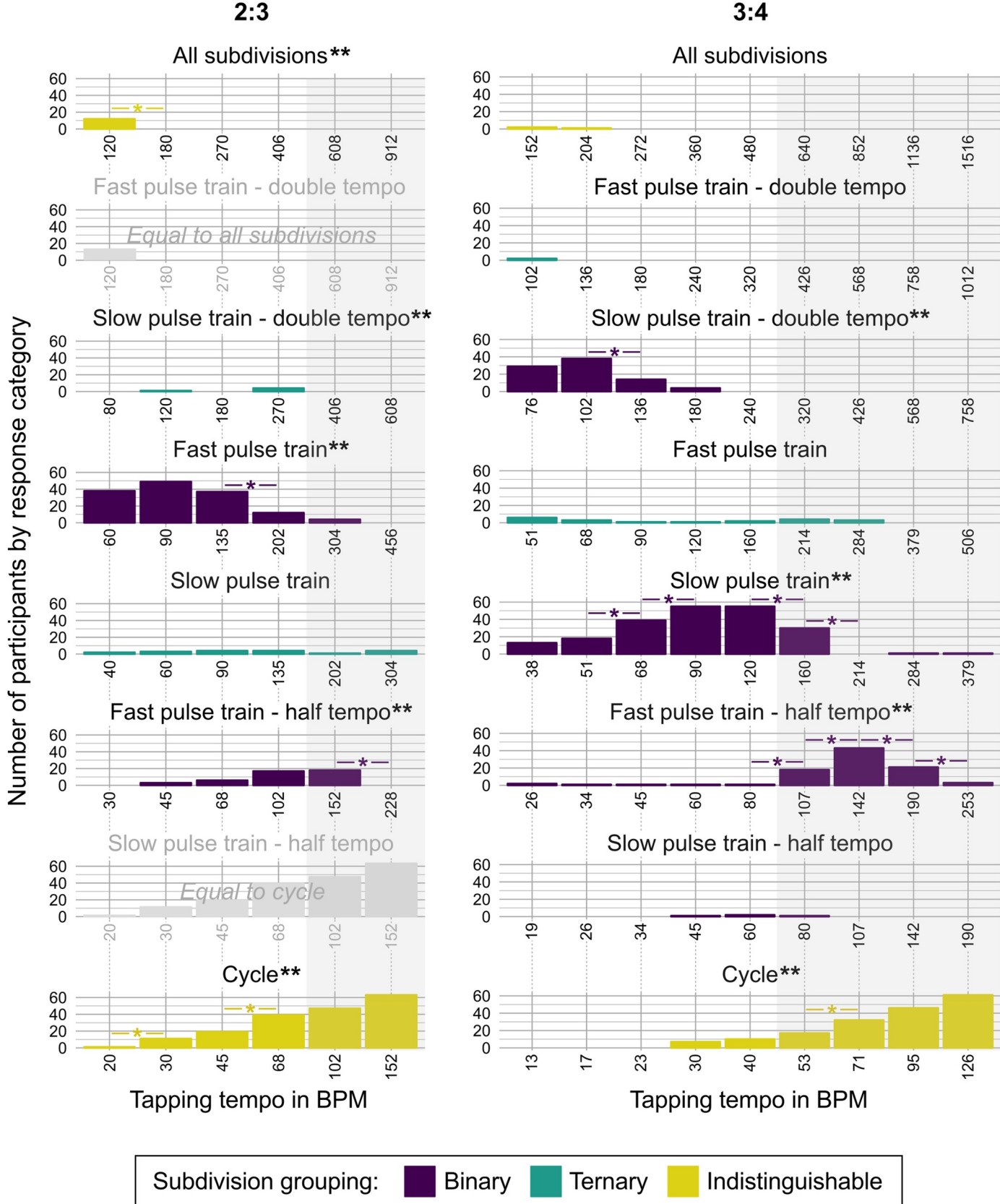

**Fig 4. Tapping behavior as a function of tempo for metrical categories of 2:3 and 3:4 polyrhythms.** Bars represent the number of participants tapping in time with each of the eight different metrical categories ranging from the fastest category, *all subdivisions* (top) to the slowest category, *cycle* (bottom). Responses to the 2:3 and the 3:4 polyrhythms are shown at different tempi ranging from slow (left) to fast (right). Individual x-axes show the tapping tempo of responses included in that metrical category. For instance, a participant listening to the slowest 2:3 polyrhythm may choose to tap in time with either *all subdivisions* at 120 BPM, the *slow pulse train—double tempo* at 80 BPM, the *fast pulse train* at 60 BPM, the *slow pulse train* at 40 BPM, the *fast pulse train—half tempo* at 30 BPM, or the *cycle* level at 20 BPM. The grey-shaded areas mark stimuli with tempi at which the subdivision tempo exceeds 600 BPM / 100 ms, i.e., the cognitive limit of subjective grouping of sounds [11, 34]. Double asterisks (\*\*) mark significant Cochran's Q tests comparing responses over different tempi within each metrical category. Asterisks (\*) mark significant McNemar's tests for pairwise comparisons between neighboring tempi within each metrical category. All tests are Bonferroni-corrected for multiple comparisons.

In 2:3 polyrhythms, most participants chose the *fast pulse train* at slower and moderate tempi (binary, slower than 135 BPM / 444 ms tap period, 270 BPM / 222 ms subdivisions). At faster tempi, most participants tapped in time with the *cycle* (indistinguishable, faster than 68 BPM / 882 ms tap period) instead of switching to the *slow pulse train* (ternary, faster than 135 BPM / 444 ms tap period, 406 BPM / 148 ms subdivisions). Notably, the second most popular response at faster tempi was to group two cycles to form a 4:6 polyrhythm and to tap every other beat of the *fast pulse train* (binary, faster than 102 BPM / 588 ms tap period, 204 BPM / 294 ms subdivisions), effectively treating the *fast pulse train* itself as the subdivision level [42]. We interpret this response, denoted *fast pulse train—half tempo* in Fig 4, as the participants' successful attempt to ensure the maintenance of a metrical structure containing binary subdivision grouping.

In 3:4 polyrhythms, most participants tapped the *slow pulse train—double tempo* at slower tempi (binary, slower than 102 BPM / 588 ms tap period, 204 BPM / 294 ms subdivisions) and switched to *slow pulse train* at intermediate tempi up to 160 BPM (binary, faster than 68 BPM / 882 ms tap period, 272 BPM / 220 ms subdivisions). Both of these responses indicate that participants perceived beats with binary subdivision grouping. At tempi faster than 160 BPM, most participants tapped the *fast pulse train—half tempo* (binary, faster than 107 BPM / 560 ms tap period, 214 BPM / 280 ms subdivisions). As in the 2:3 polyrhythm, tapping every other event in the *fast pulse train* of the 3:4 polyrhythm ensured the maintenance of a metrical structure containing binary subdivision grouping.

As expected, in both polyrhythms, *cycle* (indistinguishable) tapping increased with tempo and was the preferred response at the fastest tempi, consistent with previous reports of a preference for higher metrical levels at faster tempi [42, 44]. We also expected tapping in time with *all subdivisions* (indistinguishable) in the slowest end of the tempo range. This hypothesis was only confirmed for the 2:3 polyrhythm ($X^2(5) = 60.00$, $p < .001$, $\eta^2_Q = 0.10$). Even the slowest *all subdivisions* tempi in the experiment were quite fast (120 and 152 BPM in the 2:3 and 3:4 polyrhythms), which likely explains why only few participants preferred to synchronize with the subdivision level, especially in the 3:4 polyrhythm.

It is particularly interesting to inspect responses to the polyrhythms with subdivision tempi of approximately 100 ms / 600 BPM (see Fig 4, areas highlighted in grey; the tempi in BPM of *all subdivisions* are in the top row), as this tempo marks the lower limit of subjective grouping of sounds [11, 34]. It is reasonable to assume that a shift in response pattern around this location reflects a change in the perceived metrical structure that occurs as a consequence of the fact that grouping of subdivisions is no longer possible. In other words, the pulse train itself becomes the subdivision level when subdivisions are too fast to be subjectively grouped. Participants discarded the most common tapping preference, i.e., *fast pulse train* in 2:3 (binary, 304 BPM / 198 ms tap period, 608 BPM / 99 ms subdivisions) and *slow pulse train* in 3:4 (binary, 160 BPM / 375 ms tap period, 640 BPM / 94 ms subdivisions) around this point. In the 3:4 polyrhythm, *fast pulse train—half tempo* became the preferred response, indicating that the fast tempo caused participants to group the elements of the pulse trains rather than their

subdivisions, i.e., tap every other beat of the 4-pulse train. Furthermore, a significant shift was observed from *fast pulse train—half tempo* (binary, 253 BPM / 237 ms tap period, 506 BPM / 119 ms subdivisions) to *cycle* (indistinguishable, 126 BPM / 476 ms tap period) at the fastest tempo of the 3:4, where even the pulse train (at 506 BPM / 119 ms) approached the cognitive limit of subjective grouping of short sounds.

The distribution of tapping preferences in the present study contrasts with results of Handel and Oshinsky [31], who argued for a general preference for faster pulse trains at slower presentation rates and vice versa. They reported that the 4-pulse train of a 3:4 polyrhythm was preferred in cycle durations from 1.4 to 2.4 seconds (with 2.0 seconds corresponding to 90:120 BPM / 677:500 ms). In our study, this tempo range represented the peak of 3-pulse train responses in the 3:4 polyrhythm, i.e. *slow pulse train* (binary). Our results also differ from Handel and Oshinsky's findings for 2:3 polyrhythms. Handel and Oshinsky show an increase in the 2-pulse train and a decrease in the 3-pulse train at faster tempi. We see a similar decrease in 3-pulse tapping (binary); however, this decrease was only in favor of *cycle* (indistinguishable) and *fast pulse train—half tempo* tapping (binary), and not in favor of 2-pulse train tapping (ternary). The reasons for the discrepancies between our findings and the ones of Handel and Oshinsky are not immediately obvious. One reason could be that the categorization of responses differed in the two studies. While Handel and Oshinsky combined responses where participants tapped every other element of a pulse train with the corresponding pulse train responses, such "skipping-a-beat" responses were assigned a separate category in the present study. Another cause of the discrepancies could be differences in the stimuli. In the stimuli used by Handel and Oshinsky, the duration of individual elements in the pulse trains was always equal to half the duration of the IOIs, such that pulse trains with fewer elements consisted of longer sounds. In comparison, all pulse trains in the present study were composed of the same short cowbell sound to avoid stimulus duration confounds. It is possible that the resulting binary structure of all pulse train elements in the stimuli used by Handel and Oshinsky has weakened the propensity towards binary subdivision of the beat.

One advantage of using polyrhythms to investigate beat perception is the possibility to assess the effect of metrical structure and tempo at the same time. Our findings of the Tempo Experiment show that participants' responses followed the common distributions of preferred motor tempo around 80–160 BPM / 750–375 ms [52]. However, it is important to note that if preferred motor tempo was the only factor driving the results, we would see similar distributions of tapping responses in the slow and fast pulse trains, with the *fast pulse train* reaching its peak slightly earlier, i.e., at slightly slower stimulus tempi, than the *slow pulse train*, in both the 2:3 and the 3:4 polyrhythm. This is not the case. Rather, the tapping responses indicate a clear preference for tapping to a beat with binary rather than ternary subdivision grouping. This preference for simpler subdivision grouping is further explored in the Ratio Experiment.

## Ratio experiment

Consistent with our hypothesis, results reflected tapping preferences for pulse trains with simpler subdivision groupings, regardless of whether this was the faster or the slower pulse train in the polyrhythm. Binary subdivision grouping was preferred over ternary grouping and ternary grouping was preferred over irregular grouping. A Wilcoxon test for paired samples indicated a significant difference between tapping to polyrhythms with simpler subdivision grouping in the *slow pulse train* (3:4, 5:6; Mdn = 0.0, IQR = 0.25) and polyrhythms with simpler subdivision grouping in the *fast pulse train* (2:3, 2:5, 3:5, 4:5; Mdn = 0.83, IQR = 0.24; $Z = -7.96$, $p < .001$, $r = .73$; Fig 5B). As shown in Fig 5A, the propensity towards simpler

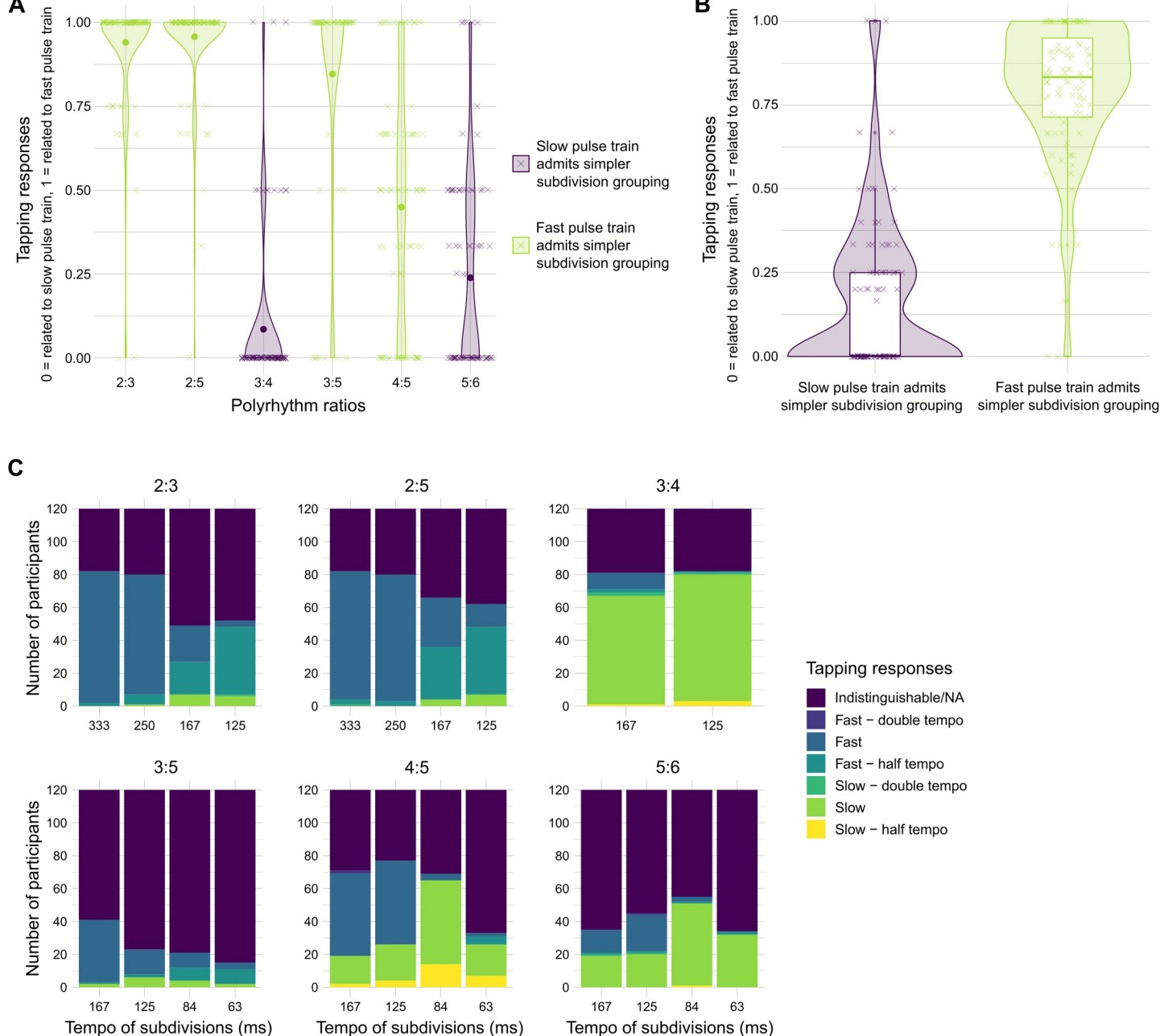

**Fig 5. Tapping responses of the experiment presenting polyrhythms of different ratios.** A) The dependent variable coded preferences for tapping in relation to the slow (0) or fast (1) pulse train. Mean across tempi are shown per participant (×) and as grand means for each ratio (•). Tapping preferences were expected to be related to the slow pulse train in polyrhythms that admit simpler subdivision in the slow pulse train (dark purple) and to the fast pulse train in polyrhythms that admit simpler subdivision in the fast pulse train (bright green). B) Mean per participant for polyrhythms that admit simpler subdivision in the slow pulse train (3:4, 5:6) in dark purple and polyrhythms that admit simpler subdivision in the fast pulse train (2:3, 2:5, 3:5, 4:5) in bright green. C) Metrical categories of the tapping responses for the individual tempi in each of the polyrhythm ratios. The category "Indistinguishable /NA" includes consistent tapping to the cycle and subdivision levels and to the polyrhythm itself as well as inconsistent and no tapping.

subdivisions is particularly strong in the four simpler polyrhythms, i.e., 2:3, 2:5, 3:4, and 3:5, whereas the 4:5 and 5:6 polyrhythms appear to be more perceptually ambiguous.

A more nuanced picture emerges when the metrical categories of the tapping responses for the individual tempi in each of the polyrhythm are taken into account (Fig 5C). The results of

the 3:4 (binary:ternary) polyrhythm show a clear preference for the *slow pulse train* (binary). As expected, this preference was flipped in favor of the *fast pulse train* of the 2:3 (ternary: binary), 2:5 (irregular:binary), and 3:5 (irregular:ternary) polyrhythms, mirroring the shift to simpler subdivision grouping in the fast pulse trains. In these polyrhythms, only seven or less participants synchronized with the slower pulse trains (ternary or irregular). Notably, increasing the tempo within each of these polyrhythms was not associated with a clear increase in *slow pulse train* responses (ternary or irregular), as a pure effect of tempo would predict. Rather, it resulted in an increase in tapping the *fast pulse train—half tempo* (binary), ensuring the maintenance of a metrical structure containing simpler grouped subdivisions. A comparison of response distributions in the 2:3 and the 2:5 polyrhythms also reveals that tempo induced transition points, specifically the increases in *fast pulse train—half tempo* responses (binary), are linked to the tempo of the subdivisions (shown on the x-axis of Fig 5C), not to the number of elements in the fast pulse train or to the overall cycle duration, both of which differ between the two polyrhythms.

In the 3:5 polyrhythm, only 21% of responses were categorized as being related to one of the two pulse trains or their half- and double-tempo equivalents, making this the polyrhythm with the least consistent pulse train tapping. This may be due to the fact that, compared to the other ratios, neither of the pulse trains in the 3:5 (irregular:ternary) polyrhythm admits a binary subdivision. At the slowest tempo, all but two of the 41 participants who engaged in consistent tapping synchronized with the *fast pulse train* (ternary, 120 BPM / 500 ms tap period, 360 BPM / 167 ms subdivisions) which admits simpler subdivision than the *slow pulse train* (irregular, 72 BPM / 835 ms tap period, 360 BPM / 167 ms subdivisions). With increases in tempo, the number of consistent responses decreased and the proportion of *fast half tempo* (binary) responses increased. *Fast half tempo* tapping (binary) indicates that participants grouped two cycles of the polyrhythm and skipped every other beat of the repeated 5-pulse train to construct an additional metrical structure. This task is more complex compared to *fast half tempo* tapping in a 2:5 polyrhythm (binary), which maintains a metrical structure with binary subdivision grouping. The new metrical structure in 3:5 is likely preferred over the two metrical structures already inherent in a single 3:5 polyrhythm cycle, because it admits a binary subdivision of the beat.

In the 4:5 polyrhythm with subdivision tempi of 167 ms and 125 ms most participants engaged in *fast pulse train* tapping (binary, 90 BPM / 668 ms tap period, 360 BPM / 167 ms subdivisions and binary, 120 BPM / 500 ms tap period, 480 BPM / 125 ms subdivisions, respectively), which is consistent with our hypothesis. However, the number of participants who preferred the pulse train with more complex grouping of subdivisions was larger compared to the polyrhythms described above. With subdivisions shorter than 84 ms, a clear preference shift was observed in favor of the *slow pulse train*. We propose two possible explanations for this shift. The first is that at these faster tempi, pulse trains are perceived as subdivisions, not because the tempo of the *fast pulse train* is too fast to tap, but because the tempo at the subdivision level (84 and 63 ms) is too fast to allow the grouping necessary to establish a sense of a beat. The second explanation is that, since the second beat of the *fast pulse train* precedes the second beat of the *slow pulse train* by only 84 ms and 63 ms respectively, the *fast pulse train* may serve as an up-beat to the *slow pulse train* which in turn becomes the perceived beat [53, 54], see also the iambic-trochaic law [55]. With the present tapping data, we cannot determine the extent to which one or both of these explanations apply.

In the slower tempi of the 5:6 polyrhythm, results showed no preference for either of the pulse trains. Responses to the faster tempi of the 5:6 polyrhythm, however, mirrored responses to the faster tempi of the 4:5 polyrhythm. This result suggests that subdivision grouping does not determine beat perception at the faster tempi of these two polyrhythms. This may be

explained by the fact that the metrical structures inherent in the 4:5 (irregular:binary) and 5:6 (binary/ternary:irregular) polyrhythms are more complex. In both polyrhythms, the two constituent pulse trains contain larger groups of subdivisions ($\geq 4$), and consequently, the subdivision levels are represented by a lower proportion of physical elements (see Table I in S1 File). These findings suggest a limit for subdivision grouping as predictor for beat perception and indicate that the propensity towards simpler subdivision grouping may only apply in cases where the auditory stimulus induces a strong feeling of a beat. This is less likely in complex metrical structures containing large groups of subdivisions.

In sum, our findings of the Ratio Experiment demonstrate that neither the number of elements in a pulse train nor the overall cycle duration determines the point at which participants switch tapping strategy. Rather, the transition points are tightly linked to the tempo of the subdivisions, and the most salient beat is represented by the pulse train with the simplest grouping of subdivisions. However, for more complex ratios, such as 4:5 and 5:6, this subdivision effect is less clear.

### Pitch experiment

As hypothesized, our results showed that pulse trains with lower pitch were more likely to be perceived as the beat. This was indicated by two individual Wilcoxon tests with significant effects of *pitch* in 2:3 and 3:4 polyrhythms ($Z = -4.98$, $p < .001$, $r = .56$; $Z = -5.52$, $p < .001$, $r = .62$, respectively; Fig 6). Comparable to the previous two experiments, participants preferred to tap in time with the pulse train that admits a binary subdivision, i.e., the *fast pulse train* in 2:3 polyrhythms and the *slow pulse train* in 3:4 polyrhythms. Attenuating the loudness of these pulse trains did not significantly affect these preferences. Fig 6 shows that participants tapped more consistently with the pulse train that admits binary subdivision, especially when the lower pitch was in this pulse train. When the lower pitch was in the pulse train admitting ternary subdivision, i.e., *slow* in 2:3 and *fast* in 3:4 polyrhythms, participants' tapping responses were split. Our interpretation is that participants either chose to tap in time with the pulse train that admits binary subdivision, or to tap in time with the lower pitched pulse train. The preference for synchronizing with the lower pitched pulse train is in line with findings from studies showing that when separating the pulse trains of polyrhythms by a musical fourth, the beat is more likely to be perceived in the lower pitched pulse train [29–31]. However, the stimuli in these studies were composed of 440 and 586 Hz pulse trains, whereas the current study used an interval of two octaves with peak frequencies of 262 and 1047 Hz. With the increased interval and the low-pitched pulse train reaching frequencies in the bass range, our stimuli more clearly speak for an effect of bass superiority.

Two individual Kruskal-Wallis tests revealed no significant effect of *loudness* in 2:3 and 3:4 polyrhythms, respectively ($X^2 = 1.32$, $p = .516$ and $X^2 = 3.93$, $p = .140$). However, the control stimuli with equal pitch but the same *loudness* manipulation as the main stimuli revealed a significant effect of *loudness* (Fig VI in S1 File). We conclude that in the current paradigm, beat perception is more strongly affected by the combined preferences for lower pitch and simpler subdivisions than by the loudness of the individual pulse trains.

Taken together, our participants preferred to synchronize movements with polyrhythm pulse trains that consist of lower pitched sounds and simpler groupings of subdivisions. Compared to these effects of low pitch and subdivisions, subtle to moderately strong loudness differences between pulse trains seem to have a smaller effect on beat preferences. Whether pitch or subdivision grouping has a stronger effect on beat perception in polyrhythms likely depends on individual factors, such as taste and familiarity, as well as on physical attributes of the stimulus, such as the size of the pitch interval between the two pulse trains, their peak frequencies,

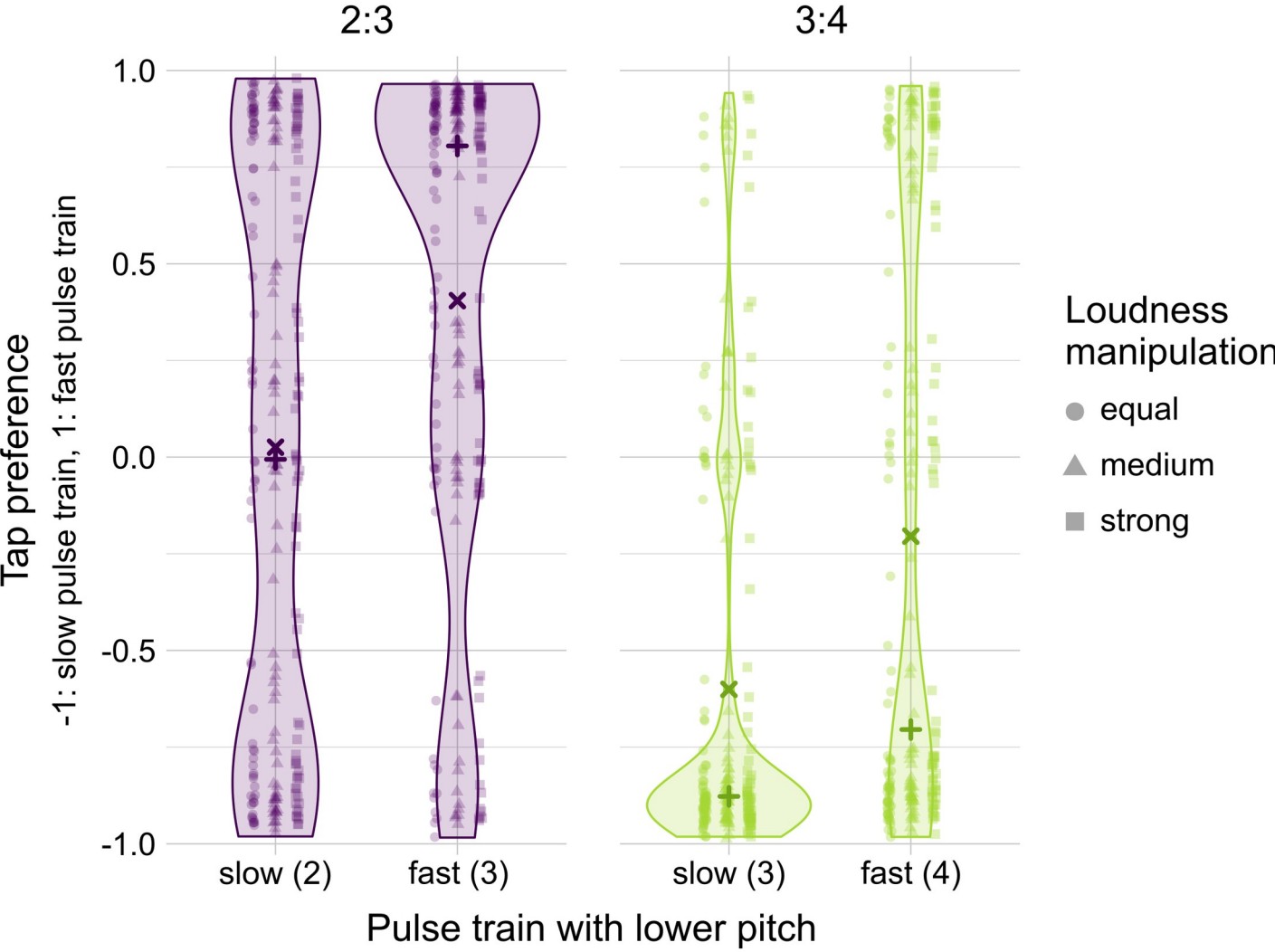

**Fig 6. Tapping responses of the experiment manipulating the pitch of individual pulse trains.** Individual data points per participant, *pitch*, and *loudness* condition of tapping preferences with 2:3 and 3:4 polyrhythms. Medians (+) and means (x) are computed over all *loudness* conditions. Values close to -1 indicate that participants consistently tapped in time with the slow pulse train, whereas values close to 1 indicate that participants consistently tapped in time with the fast pulse train.

and the ratio of the polyrhythm [32]. Physiological not mutually exclusive factors that drive the bass superiority effect include tactile stimulation [48], vestibular stimulation [56], and encoding in the auditory pathway [46]. Other factors influencing the close connection between bass, beat, and movement might be learned by exposure to music with important rhythmic information produced by bass instruments. Whether other factors promoting the bass superiority effect are innate and potentially driven by evolutionary pressures remains an open question.

## General discussion

In three online experiments (Tempo, Ratio, and Pitch), we investigated beat perception by using polyrhythms and demonstrated a yet neglected compelling influence of beat subdivisions on beat perception. Specifically, participants preferred to tap in time with beats that admit binary as compared to ternary or irregular subdivision. We showed that this preference for

binary subdivision of the beat is stable across various polyrhythmic ratios and a wide range of tempi, but influenced by a preference for low pitch. Our findings highlight the importance of metrical structures, which need to be taken into account in order to understand beat perception—an insight that has been overlooked in previous polyrhythm studies which mostly focused on characteristics of the constituent pulse trains [20, 26, 29, 32, 57, 58]. In contrast to studies focusing only on pulse train and cycle tapping responses, we focused on subdivision grouping and analyzed tapping responses at half, original, and double tempo of the polyrhythms' pulse trains to account for all levels of the competing metrical structures. As such, our findings refine the understanding of temporal grouping principles in rhythm perception by suggesting that we lean towards simplicity when grouping subdivisions, the most basic units within metrical structures of music.

We propose that subdivisions are the basic unit of beat perception and that the principle underlying the grouping of subdivisions reflects a propensity towards simplicity. Preferences for structuring time in binary units are not only found in polyrhythms, but also in more general aspects of timing in our daily lives. Binary rhythms are pervasive in human behavior and perception, such as bipedal walking [59, 60] and the "tick-tock" effect [3, 7]. Preferences for binary groupings of beats appear in early stages of development [8] and in adults [61], and simpler grouping of beats are statistical universals across musical cultures [62]. A binary categorization of events emerges even spontaneously when irregular rhythms are imitated from participant to participant in iterated learning paradigms [63], likely due to an automatic subdivision of the beat into two intervals [37]. This interpretation is supported by the finding that sensorimotor synchronization to a periodic beat is generally more consistent when it is subdivided into two events, compared to three, four or five events; an advantage that can be observed in the brain when measuring the steady-state evoked potentials related to duplets [64]. Savage and colleagues [62] listed the presence of "2- or 3-beat subdivisions" as the first statistical universal in a sample of 304 cross-cultural music recordings. While this description appears to pertain to the subdivision level, closer inspection reveals that it does in fact mean subdividing the cycle into two or three beats. Consequently, this statistical universal holds no information as to whether these beats admit binary or ternary subdivision. This is an example of how inconsistent uses of the terminology describing grouping processes at different metrical levels can complicate direct comparisons between studies. Nonetheless, the propensity towards simplicity shown in all these studies align well with our interpretation that beat perception follows a subdivision hierarchy in favor of binary groupings, the simplest organization of temporal information.

Our findings are consistent with the general conclusion drawn from the seminal studies on polyrhythm perception by Handel and colleagues: Rhythmic interpretation is contextual and depends on a number of factors, including tempo, ratio, relative intensity, and pitch [29–31, 33]. However, we do not agree with Handel's notion that "There are no basic units of rhythm, and no theories based on grouping principles or tree representations seem able to encompass rhythmic diversity" [29, p. 481]. Our Tempo Experiment showed that binary grouping of subdivisions was preferred regardless of whether the beat corresponded to the fast or the slow pulse train (Figs 3 and 4). Thus, a principle of simpler grouping pertains to the subdivisions, not the beats, making the subdivision the basic unit of rhythm. Our Ratio Experiment expanded the validity of this grouping principle to more complex polyrhythms and outlined its limits. When binary grouping of subdivisions was not present in the polyrhythm, as in the case of the 3:5 polyrhythm, listeners preferred ternary to irregular grouping of subdivisions or they constructed a binary subdivision level by concatenating two cycle repetitions (Fig 5). The preference for simpler subdivision grouping became less clear for our most complex polyrhythms, i.e., 4:5 and 5:6. Our Pitch Experiment showed that the preference for simpler

grouping of subdivisions decreased when the pulse train with more complex grouping of sub-divisions was presented at a lower pitch (Fig 6). The 6 and 12 LKFS loudness differences between the pulse trains had a surprisingly small effect on beat preferences compared to the preference for lower pitched pulse trains and for grouping subdivisions in pairs. Therefore, our three experiments support the notion that rhythmic interpretation is contextual. Importantly, our findings demonstrate that we have to extend the list of contextual factors to include grouping at the subdivision level as a critical determinant of beat perception in polyrhythms.

The finding that people tap in a way that reflects the simplest possible grouping of subdivisions may explain discrepancies between findings in previous beat perception studies. For instance, using real musical excerpts, McKinney and Moelants [19] observed that the most saliently perceived tempo for some excerpts deviated largely from the preferred tempo of 120 BPM [44, 58]. Our results showed that tapping preference transition points were tightly linked to the tempo of the subdivisions, rather than to the tempo of the pulse trains. Similarly, temporal preferences in beat perception in real music may not be constrained by the tempo of the beat itself, but by the tempo and grouping of the underlying subdivisions.

A growing body of research suggests that experience within a musical culture shapes perception and consistency of production of musical rhythms prevalent in that culture [65, 66]. A pivotal question concerns the universality of the proposed preference for binary subdivision grouping, considering the large cross-cultural variation in music traditions. Our study sample consisted mainly of Western participants (Western Europe, North America). Simple binary rhythms are dominating the Western musical culture whereas more multifaceted rhythmic patterns are prevalent in non-Western musical cultures, e.g., in the Balkans, Africa and Latin America [for an overview see e.g., 67]. Not only do West African drummers exhibit considerable rhythmic flexibility in terms of switching between metrical levels of the typical 12-unit rhythmic patterns, they are also more likely to perceive four regular beats, i.e., with ternary subdivision of the beat [68]. In contrast, the participants in the present study perceived three regular beats with four subdivisions in the comparable 12-unit 3:4 polyrhythm, and avoided synchronizing with the fast pulse train that admitted a ternary subdivision. The lack of cultural variation in our participant sample is clearly a limitation. However, a recent cross-cultural study shows that perceptual priors representing small-integer ratio rhythm categories are universal [69]. Given the universality of simple subdivision grouping, a remaining key question is whether the contrasting beat perception in the two populations reflects differences in metrical flexibility. In other words, the simplest grouping patterns are present even in musical cultures with rich, multifaceted rhythmic patterns, and members of these cultures are likely more rhythmically adept. Importantly, musical training is also associated with greater metrical flexibility as indicated by studies showing that trained musicians demonstrate more complete hierarchical representations of the rhythmic structure of music [70–72]. Perhaps the lack of metrical flexibility constrains the tapping responses of Western listeners to correspond to the simplest possible grouping of subdivisions, whereas e.g., West African listeners can more easily switch between metrical levels [68].

Claims of universality of a perceptual phenomenon can be supported by neurophysiological evidence. The current study did not collect neurophysiological data and cannot directly discuss the universality of the results. Yet, recent evidence for the presence of oscillatory dynamics in auditory cortex during rhythm processing align well with our findings of a preference for simple subdivision groupings [73]. Neural resonance models that describe the brain as a system of coupled non-linear oscillators predict more power at frequencies closer to the stimulus frequency, i.e., in our context more power at frequencies related to binary compared to ternary subdivisions [74]. Such higher-order resonance could explain why the peak was larger at the frequency related to the 3-beat (binary) than the 2-beat (ternary) in the 2:3 polyrhythm

presented to participants in a study by Chemin and colleagues [75]. Overall, higher-order resonance phenomena in the brain provide neurophysiological explanations for the preference for 1:2 over 1:3 temporal relationships [74], which in turn may explain why polyrhythms are not perceptually ambiguous.

The knowledge about the universality and limits of the human propensity towards simple subdivision grouping in temporal processing still has to be refined. To this aim, future research should assess the generalizability and exceptions of binary subdivision preferences in beat perception in real music, sensorimotor synchronization with rhythms perceived in different modalities, and timing in social interactions involving movement, dance, music or language. In future studies, we will expand the current findings in a number of exploratory analyses mentioned in the preregistration for the present study (https://aspredicted.org/yi5si.pdf). We will examine the influences of musical training, individual spontaneous motor tempo, native languages, and cultural background on subdivision grouping preferences in rhythm processing.

Studying the hierarchical organization of temporal patterns is an interdisciplinary effort. From the perspective of music perception and cognition, our findings reveal the importance of lower hierarchical levels, such as subdivisions, to grouping processes at higher levels, such as beat and meter. Our study adds to evidence suggesting that in everyday behavior, time is structured in binary units.

## Supporting information

**S1 File. The file contains the following table and figures.** Table I. Polyrhythm Complexity; Fig I. Examples of Subdivision Grouping and Tapping Responses in a 2:3 Polyrhythm; Fig II. Data Cleaning and Circular Analyses to Categorize the Tapping Responses; Fig III. Tapping Data Overview; Fig IV. Tempo Control Trials with 120 BPM; Fig V. Tempo Control Trials at Faster Tempi; Fig VI. Pitch Control Trials.
(PDF)

## Author Contributions

**Conceptualization:** Cecilie Møller, Jan Stupacher, Alexandre Celma-Miralles, Peter Vuust.

**Data curation:** Cecilie Møller, Jan Stupacher, Alexandre Celma-Miralles.

**Formal analysis:** Cecilie Møller, Jan Stupacher, Alexandre Celma-Miralles.

**Methodology:** Cecilie Møller, Jan Stupacher, Alexandre Celma-Miralles, Peter Vuust.

**Writing – original draft:** Cecilie Møller, Jan Stupacher, Alexandre Celma-Miralles.

**Writing – review & editing:** Peter Vuust.

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
