## [Decision Letter · Decision Letter 0]

28 May 2021

PONE-D-21-15256

Beat Perception in Polyrhythms: Time is Structured in Binary Units

PLOS ONE

Dear Dr. Møller,

Thank you for submitting your manuscript to PLOS ONE. After careful consideration, we feel that it has merit but does not fully meet PLOS ONE’s publication criteria as it currently stands. Therefore, we invite you to submit a revised version of the manuscript that addresses the points raised during the review process.

Overall, all three reviewers were extremely positive about the science conducted. They do have suggestions that will substantially improve the paper, with which I concur. It's clear that the terminology, despite best attempts using a clarifying figure, becomes confusing and difficult to follow over the course of the manuscript. All reviewers make suggestions about specific phrases that were confusing and how consistent definitions and use of terminology would improve the communication of the concepts. In addition, there are some discussion points that deserve consideration (e.g., binary preference in non-Western rhythms) and methodological clarifications. I am confident that these issues can be addressed in a revision.

We look forward to receiving your revised manuscript.

Kind regards,

Jessica Adrienne Grahn

Academic Editor

PLOS ONE

Journal Requirements:

Reviewers' comments:

Reviewer's Responses to Questions

**Comments to the Author**

1. Is the manuscript technically sound, and do the data support the conclusions?

Reviewer #1: Yes

Reviewer #2: Partly

Reviewer #3: Yes

2. Has the statistical analysis been performed appropriately and rigorously? 

Reviewer #1: Yes

Reviewer #2: Yes

Reviewer #3: Yes

3. Have the authors made all data underlying the findings in their manuscript fully available?

Reviewer #1: No

Reviewer #2: Yes

Reviewer #3: Yes

4. Is the manuscript presented in an intelligible fashion and written in standard English?

Reviewer #1: Yes

Reviewer #2: Yes

Reviewer #3: Yes

5. Review Comments to the Author

Reviewer #1: In this manuscript, the authors deploy a massively crowdsourced tapping study to polyrhythmic auditory stimuli, and use the results to argue that subdivision structure is a key determining factor in where human listeners perceive a beat and tap along. Specifically, they claim that listeners are strongly biased toward identifying beat structures such that events between beats subdivide the beat into two rather than three or more even parts.

The argument is clear, cohesive, and well supported by the data. It adds an important and often overlooked dimension to the discussion of beat perception. The hypothesis and the insightful discussion around it are well articulated and seem to be profoundly informed by strong musical intuition. Overall, I believe it is a valuable contribution to the field that will help shape the discussion moving forward.

The formidable challenge these authors faced in communicating these results is navigating the many pieces of information necessary to fully characterize each tapping pattern — tempo, subdivision level, subdivision duration, whether it the tap is the faster or slower click track or a doubled or halved version of one of them. In Figure 1, they do an excellent job defining useful terms, which they use consistently throughout the paper. However, as they discuss their results in the text and display them in figures, it sometimes becomes difficult for the reader to map the specific tap pattern categories onto the take-away messages about subdivision. I would like to see the authors work to communicate their key message more clearly through language and image. Here are some example suggestions:

* At some points in the text, they refer to the “slow pulse train” and then give the subdivision level (binary/ternary) in parentheses — I would strongly encourage the authors to use such a convention systematically throughout, and to go even further when appropriate by specifying, e.g., (ternary, 480ms tap period, 120ms subdivisions). I recognize that this could lead to clutter, but I think it’s at least worth taking another step in this direction in the Results section.

* Looking over Figure 4, it is not immediately clear which tap patterns are associated with binary and which with ternary subdivisions. Perhaps a color coding convention would help, or at least careful labeling, to highlight the tendency toward binary subdivision, which is not readily observable in this figure without much careful interpretation.

* The language in Figure 5 is confusing — “tapping expected to be related to the slow pulse train” is a strange way of saying “slow pulse train admits binary subdivision"

Additional concerns:

The number of trials excluded is pretty substantial. I would like to see some discussion of what was going on in those cases, or at least assurance that the exclusion process is not skewing the data.

I find it a little confusing that the authors lump all even subdivision patterns in with binary subdivision. I would appreciate a little more clarity in the discussions of tapping patterns that have 4, 6, etc. subdivisions —they call these “binary subdivision groupings,” but the 6 group could equally be called ternary. Perhaps language like “admit a binary subdivision” might be more precise?

Finally, the data were not actually available to me — it seems that they were password protected, and I did not have a “review key” with which to view them.

Reviewer #2: The experiments per se were well performed and analyzed. However, see my main concern below, some concepts are confusing, and the rationale of the exact question to be addressed, as well as how the conclusion is supported, are unclear.

Main concern:

For a common basis of concepts and terminology to discuss. A meter can be subdivided into beats (e.g., a 2/4 meter or a 3/4 meter), and a beat can also be subdivided into lower levels (e.g., into 2, 3 subdivisions). The preference of binary over ternary subdivision, on both the meter or the beat level, has been studied. Given a singer and a pianist who play together, they follow the same meter, i.e., the same subdivision on the meter level; may or may not follow the same subdivision on the beat level. This is a common situation, in which they had a consistent metrical structure (here, yes, subdivision on the beat level can be different), which had the same sense of beat that allows them to play synchronously.

In the manuscript, terms such as “subdivision”, “metrical structure”, “metrical level”, are ambiguously used, in terms of meter of beat. For example, Fig. 1A, what “circle” refers to? If meter (I guess), then why, or in what kind of situation, would one play in a 2/4 meter and another plays in a 3/4 meter? In other words, why the referred “2:3” polyrhythm is a question to address? If beat, yes, people often do it, also the manuscript describes “are often used to create tension and increase expressiveness in musical performances”. But again, as above, it is common for same subdivision on the meter level and same/or different subdivision on the beat level. What is the exact question?

It seems that subdivision on beat level and subdivision on meter level are confused. The corresponding descriptions and statements are ambiguous and confusing, e.g., “a yet neglected compelling influence of beat subdivisions on beat perception”, “Our findings highlight the importance of metrical structures, which need to be taken into account in order to understand beat perception—an insight that has been overlooked in previous polyrhythm studies.The authors please explain the rationale of the study, what exactly is the question. Perhaps, previous studies did not neglect or overlook the referred question.

Reviewer #3: The authors explore beat perception in polyrhythms through three online finger tapping experiments and make an interesting claim that beat, rather than being primarily constrained by tempo (as is widely thought in the field), is primarily constrained by subdivisions, with a strong preference for binary groupings. This claim/perspective is very intriguing and important for the field. The paper is also very well written and presented. In my opinion, the work definitely deserves to be published, but I have a few concerns.

My biggest concern is the highly "Western" makeup of participants, because perhaps the conclusion that "we prefer to structure time in binary units" could be a result of the tendency of Western popular music to be in binary meters. To me, the data presented are not sufficient to conclude that we (=all humans) biologically have this binary bias. However, I appreciate this perspective proposed by the authors, so perhaps a deeper discussion of the limitations of the demographics of the sample and/or an additional analysis of the countries/languages/background of the minority of participants who preferred ternary groupings, would add some necessary nuance to the central claim of the paper. Along these lines, the cross-cultural iterated reproduction study of Jacoby & McDermott 2017 is not mentioned at all - how consistent is cross-cultural work with the idea of a universal preference for binary subdivisions?

The other aspect of this paper that needs improvement is the terminology around the different tapping patterns, which becomes increasingly confusing. For example, in Fig 3: "Tapping responses related to groupings of binary and ternary subdivisions are concatenated separately": what does this and the text that follows mean? You also say "binary grouping in 2:3 (fast pulse train including half tempo)", then also "binary grouping in 3:4 (slow pulse train, including half tempo and double tempo, and fast pulse train - half tempo) it is very challenging to follow this terminology. Fig. 1 does an excellent job of illustrating the concepts of subdivision, beat, cycle, and stimulus - perhaps fig 1 (or some version of it) could be expanded to include visually showing the tapping pattern that each of the tapping patterns in fig 4 corresponds to.

Other questions:

- how were taps collected? were they laptop key presses? how reliable were tap times? how much delay/jitter from the operating system or browser?

- the inclusion criteria for trials seems quite strict, especially if only 54.1% of trials were finally included in the Ratio experiment. Could there be higher motor error while tapping to irregular / ternary groupings, and could that cause trials to be excluded, effectively biasing data in favor of binary?

- pitch task: was the 2-octave choice deliberate? why? in this case, low and high tones are similar perceptually, in the sense that they would fuse perceptually if played at the same time. would a more dissonant/contrasting higher pitch (e.g. tritone) have been more effective at attracting synchronization?

- the discussion of Handel and Oshinsky (1981) is incomplete... why were their results were different? different demographic of participants? differences in stimuli?

- "tapping preference transition points were tightly linked to the tempo of the subdivisions, rather than to the tempo of the pulse trains" - this is a really neat and important point from the Tempo experiment, but it is not actually at all obvious from fig 3-4. I wonder if you could somehow show this in a figure. perhaps the "stacked" design of the stimuli (Table 1) where an interval that is the "3" in one stimulus is the "2" in another can be exploited to make this point?

- general discussion paragraph 2: there may also be evidence for tick-tock effect in macaques: https://papers.ssrn.com/sol3/papers.cfm?abstract_id=3790895 also, it would be helpful to describe in more detail the specific findings in Savage et al 2015 regarding binary vs. ternary meters in music across cultures, and whether it is consistent with the findings here.

- though this is a behavioral study, it is odd that there is no discussion at all on possible neurophysiological accounts for the preference for binary subdivision. could differences in beat salience of the different groupings account for the data (Rajendran, et al 2017)? Are the data consistent with tapping+EEG studies of beat perception (Nozaradan et al 2011;2012)? Would neural resonance theory predict a preference for binary groupings (Large et al 2015)?

Typos:

under "Participants", 4th line: "youth countries"? also, the percentages don't add up

under "Statistical analyses", 2nd line: fast pulse train is repeated

6. PLOS authors have the option to publish the peer review history of their article (what does this mean?). If published, this will include your full peer review and any attached files.

Reviewer #1: **Yes: **Jonathan Cannon

Reviewer #2: No

Reviewer #3: No

---

## [Author Response · Author response to Decision Letter 0]

12 Jul 2021

See attached document "Response_to_reviewers"

---

## [Decision Letter · Decision Letter 1]

2 Aug 2021

Beat Perception in Polyrhythms: Time is Structured in Binary Units

PONE-D-21-15256R1

Dear Dr. Møller,

We’re pleased to inform you that your manuscript has been judged scientifically suitable for publication and will be formally accepted for publication once it meets all outstanding technical requirements.

There are some minor suggestions (e.g., one missing reference) that should be minimal to incorporate.

Kind regards,

Jessica Adrienne Grahn

Academic Editor

PLOS ONE

Additional Editor Comments (optional):

Reviewers' comments:

Reviewer's Responses to Questions

**Comments to the Author**

1. If the authors have adequately addressed your comments raised in a previous round of review and you feel that this manuscript is now acceptable for publication, you may indicate that here to bypass the “Comments to the Author” section, enter your conflict of interest statement in the “Confidential to Editor” section, and submit your "Accept" recommendation.

Reviewer #1: All comments have been addressed

Reviewer #2: All comments have been addressed

Reviewer #3: All comments have been addressed

2. Is the manuscript technically sound, and do the data support the conclusions?

Reviewer #1: Yes

Reviewer #2: Yes

Reviewer #3: Yes

3. Has the statistical analysis been performed appropriately and rigorously? 

Reviewer #1: Yes

Reviewer #2: Yes

Reviewer #3: Yes

4. Have the authors made all data underlying the findings in their manuscript fully available?

Reviewer #1: Yes

Reviewer #2: (No Response)

Reviewer #3: Yes

5. Is the manuscript presented in an intelligible fashion and written in standard English?

Reviewer #1: Yes

Reviewer #2: Yes

Reviewer #3: Yes

6. Review Comments to the Author

Reviewer #1: Authors gave thoughtful and thorough responses were given to all reviewer comments and made the paper clearer in various ways. I have no further criticisms!

Reviewer #2: As I have said, the experiments were well conducted; I therefore recommend for publication in PlosOne.

In addition, below are suggestions, provided the authors would like make their work more clear. For my concern, the authors have made some clarifications. To be simple, for Figure S1; left: ternary subdivision on beat level whereas binary subdivision on meter level; right: binary subdivision on beat level whereas ternary subdivision on meter level. This type of description would be clearer, on which “metrical level” “binary/ternary subdivision” refers to, and what exactly the “metrical structure” refers to. Given such clarifications (and thereafter clarification with existing literature), as I concerned, I hope the authors could make the theoretical significance of their work further clearer.

Reviewer #3: The authors have addressed all of my concerns. My only minor suggestion is to reconsider the use of the word "admit" throughout the manuscript, which to me sounds strange/wrong. Some examples below (page numbers refer to the manuscript with tracked changes):

p. 9: "admits" doesn't seem like the right word...

The fast pulse train in the 2:3 polyrhythm (i.e., 3) admits binary subdivision grouping, whereas the slow pulse train in the 3:4 polyrhythm (i.e., 3) admits binary subdivision.

Suggestion: Onsets of the fast pulse train in the 2:3 polyrhythm (i.e., 3) mark binary subdivision, as do onsets of the slow pulse train in the 3:4 polyrhythm (i.e., 3).

p. 13: the word "admits" again... consider replacing with

For statistical analysis, we computed two means: 1) the mean of all polyrhythm ratios where the simpler subdivision grouping was in the faster pulse train (i.e. 2:3, 2:5, 4:5, 3:5), and 2) the mean of all polyrhythm ratios where the simpler subdivision grouping was in the slower pulse train (i.e. 3:4, 5:6).

p. 16: maybe replace the word "admitting" with "representing"?

p. 29: "volitional control (ref)" - ref missing

7. PLOS authors have the option to publish the peer review history of their article (what does this mean?). If published, this will include your full peer review and any attached files.

Reviewer #1: **Yes: **Jonathan Cannon

Reviewer #2: **Yes: **Xiang Wu

Reviewer #3: No

---

## [Editor Report · Acceptance letter]

11 Aug 2021

PONE-D-21-15256R1 

Beat Perception in Polyrhythms: Time is Structured in Binary Units 

Dear Dr. Møller:

I'm pleased to inform you that your manuscript has been deemed suitable for publication in PLOS ONE. Congratulations! Your manuscript is now with our production department. 

Kind regards, 

on behalf of

Dr Jessica Adrienne Grahn 

Academic Editor

PLOS ONE